# Attosecond nanoscale near-field sampling

B. Förg[1,2], J. Schötz[1,2], F. Süßmann[1,2], M. Förster[1,3], M. Krüger[1,3], B. Ahn[4,5], W.A. Okell[1], K. Wintersperger[1], S. Zherebtsov[1,2], A. Guggenmos[1,2], V. Pervak[2], A. Kessel[1], S.A. Trushin[1], A.M. Azzeer[6], M.I. Stockman[1,7], D. Kim[4,5], F. Krausz[1,2], P. Hommelhoff[1,3] & M.F. Kling[1,2]

The promise of ultrafast light-field-driven electronic nanocircuits has stimulated the development of the new research field of attosecond nanophysics. An essential prerequisite for advancing this new area is the ability to characterize optical near fields from light interaction with nanostructures, with sub-cycle resolution. Here we experimentally demonstrate attosecond near-field retrieval for a tapered gold nanowire. By comparison of the results to those obtained from noble gas experiments and trajectory simulations, the spectral response of the nanotaper near field arising from laser excitation can be extracted.

[1] Max Planck Institute of Quantum Optics, Hans-Kopfermann-Str. 1, D-85748 Garching, Germany. [2] Department of Physics, Ludwig-Maximilians-Universität München, Am Coulombwall 1, D-85748 Garching, Germany. [3] Department of Physics, Friedrich-Alexander-Universität Erlangen-Nürnberg, Staudtstraße 1, D-91058 Erlangen, Germany. [4] Department of Physics, CASTECH, POSTECH, Pohang, Kyungbuk 790-784, Republic of Korea. [5] Max Planck Center for Attosecond Science, Pohang, Kyungbuk 790-784, Republic of Korea. [6] Attosecond Science Laboratory, King-Saud University, Riyadh 11451, Saudi Arabia. [7] Department of Physics and Astronomy, Georgia State University, Atlanta, Georgia 30303, USA. Correspondence and requests for materials should be addressed to P.H. (email: peter.hommelhoff@physik.uni-erlangen.de) or to M.F.K. (email: matthias.kling@lmu.de).

Photoemission from solids is one of the most fundamental and long-studied electron phenomena in nature. Related photon–electron interactions form the basis for modern optoelectronics, where light can trigger electron transfer, amplification and emission; vice versa, electron injection and excitation can result in the emission of light[1]. The decrease of the dimensions, and increase in speed of electronic and optoelectronic circuitry is paramount for improving their performance, with switching rates possibly approaching optical frequencies in all-optical wide-bandgap devices[2–4]. This motivates the development of femtosecond[5–10] to attosecond[11,12] metrology of nanolocalized fields and the control of electron emission and acceleration in these fields[13–16]. Attosecond resolution is a prerequisite in tracing, among others, optical-field-induced formation and subsequent relaxation of collective electron dynamics, transient changes in the optoelectronic properties of nanostructured materials in strong fields and screening after photoemission.

Attosecond nanoscale near-field sampling (ANNS), proposed in 2007 (ref. 12) and extensively studied theoretically[17–22], has been shown to provide sub-cycle resolution of optical near-field dynamics in nanostructured materials, but has not yet been implemented experimentally. ANNS relies on the emission of photoelectrons with high initial momentum by an attosecond extreme ultraviolet (XUV) pulse, and subsequent acceleration of the photoelectrons in the near fields. As a linear process, XUV photoemission results in high-energy electrons being emitted from the entire illuminated area that is typically not confined to a nanoscale, but rather micron scale. Consequently, the detection scheme averages on the micron scale making it challenging to characterize near fields around nanostructured sample geometries, typically varying substantially in phase and amplitude on a nanometre scale[17–20].

Here we perform ANNS measurements on a nanotaper at near-infrared (NIR) intensities well below the onset of non-linear effects. Using the gold nanotaper sample geometry, we show that through careful analysis of field homogeneity and streaking electron trajectories, a meaningful attosecond characterization of near fields can be performed in spite of the inherent challenges associated with the large emission area.

## Results

**Experimental approach.** ANNS is based on attosecond streaking spectroscopy[23,24], where the XUV photoemitted electron is accelerated by a few-cycle laser field with a variable time delay. The change of the electron's momentum can be described by the classical equation of motion $\Delta\mathbf{p} = -e \int_{t_0}^{\infty} \mathbf{E}(\mathbf{r}(t), t)\mathrm{d}t$, where $t_0$ is the emission time, $e$ the elementary charge, $\mathbf{r}$ the electron position and $\mathbf{E}$ the electric field. In the 'ponderomotive' streaking regime, the spatial variation of the electric field during electron propagation in the laser pulse can be neglected, and the integral can be evaluated to yield the vector potential $\mathbf{A}(t_0)$ (Coulomb gauge). The final electron energy $E_{\mathrm{kin}}$ is then given by

$$E_{\mathrm{kin}} = \frac{1}{2m_e}(\mathbf{p}_0 + \Delta\mathbf{p})^2$$
$$\approx \frac{1}{2m_e}\left[p_0^2 - 2e\mathbf{p}_0 \cdot \mathbf{A}(t_0) + e^2 A^2(t_0)\right], \qquad (1)$$

where $\mathbf{p}_0$ and $m_e$ are the initial momentum and mass of the electron. For the moderate field strengths used in attosecond streaking, the last term can be neglected. The change of the electron kinetic energy is thus directly related to the vector potential in the direction of electron emission and accordingly preserves full temporal information of the probed electrical fields. For nanostructured samples, there can be several streaking

regimes[17]. The measurements presented here are in the ponderomotive regime discussed above.

Figure 1a shows our experimental set-up for ANNS. The experimental approaches are described in detail in the methods section and supporting information (Supplementary Note 1; Supplementary Fig. 1). In brief, phase-stabilized, 4.5 fs few-cycle NIR pulses (720 nm central wavelength) are focused onto a gold nanotaper. The laser field excites collective electron dynamics in the nanotaper, resulting in spatially varying near fields. Attosecond XUV pulses of 220 as duration and 95 eV central photon energy (Supplementary Fig. 2) with adjustable delay release electrons from the sample, which are subsequently accelerated in the near fields. The momentum distribution of the freed electrons is recorded as a function of delay between the XUV pulse and the NIR field by a time-of-flight (TOF) spectrometer. The spectrometer axis is aligned parallel to the laser polarization and the nanotaper axis. Delay-dependent variations of the momentum component parallel to the spectrometer axis are thereby measured. The taper can be replaced by a gas target (Ne), allowing independent characterization of the incident NIR field and the XUV pulse by means of standard attosecond streaking[24].

The nanotaper is formed by a gold nanowire with a 10° opening angle (Supplementary Fig. 3; Supplementary Note 2; the Method section). The XUV focal spot size is ~5 μm, and is centred on the end of the nanowire. Thus, the XUV probes a 2.5-μm length of the nanowire, with its diameter tapering from ~640 nm down to 200 nm. The surface area of the region of enhanced fields at the end of the hemisphere (100-nm radius of curvature) that terminates the tapered nanowire is only a small fraction (~1–2% of the total surface area), and does not yield any detectable signal above the noise level in our measurements (Supplementary Note 7). Thus, only the near fields surrounding the wire play a role in our measurements, and not those surrounding the hemisphere.

**Experimental results.** Figure 2a shows a typical experimental photoelectron spectrum obtained from the two-color interaction with the nanotaper. The spectrum reveals two major contributions: a low-energy contribution <20 eV and a broader structure between 50 and 93 eV, with its maximum ~80 eV. Low-energy electrons are mainly generated by strong-field NIR photoemission from the sharp apex region of the nanotaper with a cutoff energy of 15 eV consistent with the applied intensities of ~$1 \times 10^{11}$ W cm$^{-2}$, as has been observed in related strong-field photoemission studies on nanotips with similar apex radii[13,15,25]. Photoelectrons with an energy exceeding 50 eV are attributed to XUV photoemission. We want to point out that spatial scanning of the nanoscale target in the focal plane supports these two different photoemission schemes. Figure 2b shows localization of the strong-field NIR photoemission at the nanotaper apex due to the strong nonlinearity of the process. This is a super-resolution phenomenon, which we used for precise positioning of the nanotaper in the laser focus. In contrast, the XUV photoemission is a linear process, leading to the electron emission from the whole illuminated surface. The map in Fig. 2c, thus represents the convolution of the shape of the nanotaper with the shape of the XUV beam profile.

Figure 3a shows a streaking spectrogram obtained from the Au nanotaper. The panel to the right depicts the spectrum at a fixed delay. The high-energy edge of the spectrum is assigned to the photoemission of Au-5d electrons. Figure 3b shows the streaking spectrogram obtained from Ne for the same pulse parameters. Under the assumption that quantum effects can be neglected and negligible delay for the absolute photoemission time from Ne atoms[26], the gas streaking gives access to the vector potential of

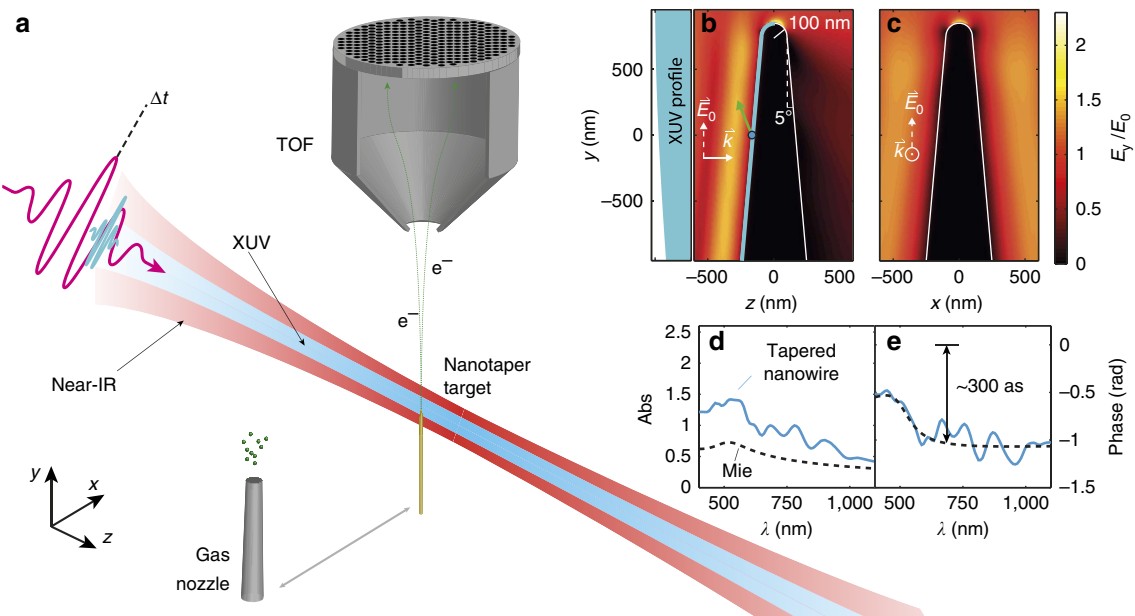

**Figure 1 | Attosecond nanoscale near-field sampling.** (**a**) Experimental set-up: few-cycle near infrared (NIR) and isolated extreme ultraviolet (XUV) attosecond pulses with variable delay are focused onto a gold nanotaper. High-energy electrons are emitted via XUV photoionization and subsequently accelerated in the local near fields. The delay-dependent final kinetic energy is measured using a time-of-flight (TOF) spectrometer. The nanotaper can be replaced by a gas target. (**b,c**) Maximal normalized local field strengths of the component parallel to the taper axis (**b**) along the laser propagation direction and (**c**) perpendicular to it as obtained from FDTD simulations. The green arrow shows the maximum electron detection angle. The blue lineout illustrates the XUV photoemission area. (**d,e**) Response function of the probed $E_y$-component (axis of laser polarization and TOF spectrometer) for a representative point (as indicated in **b**) at the nanotaper, showing the (**d**) absolute value and (**e**) phase dependence on the wavelength. The response of the shank is close to an infinite cylinder with a diameter of 200 nm calculated using Mie theory (black dashed line). Slight position-dependent oscillations occur due to a plasmon launched at the tip apex (Supplementary Note 4 and 5).

the incident laser field[27]. The extracted streaking curves from Fig. 3a,b are compared with Fig. 3c (Supplementary Note 3; Supplementary Figs 4 and 5). A shift between the streaking traces is directly discernible and is evaluated from this particular set of measurements as $\Delta t = (200 \pm 50)$ as.

Figure 4a shows the reconstructed electric fields from the measurements on the nanotaper shown in Fig. 3b. Our measurements not only allow a direct retrieval of the near field, but also allow a reconstruction of the spectrally resolved response function for the spectral range of the incident laser pulse at the sample surface (Supplementary Note 4). This is particularly useful, since in principle this allows the prediction of the near field for any synthesized light field within the same spectral range. Figure 4b,c shows the response function, calculated from experimental data, in terms of the wavelength-resolved phase shifts and relative amplitudes, respectively, between the local near field and the incident laser field under varying experimental conditions (different days, different tapers). The crosses represent data points obtained by discrete Fourier transform of the reconstructed electric fields of gas and nanotaper measurements. Solid dots show the average response, retrieved by a linear interpolation on the wavelength axis of the experimental data points that were weighted by the respective electron count rate. Error bars on the solid dots are calculated as the respective standard deviation (s.d.). The average measured phase shift lies between $-0.4$ and $-0.8$ rad.

**Theoretical results**. Figure 1b,c displays the near-field component in the detection direction, calculated numerically using a finite-difference time-domain (FDTD) approach. The near field can be understood as a superposition of incident and scattered

fields, and displays a substantial variation of field strength over length scales of 200 nm from the surface. Figure 1d shows the response function for the nanotaper, yielding a phase shift of the near fields with respect to the laser pulse. The phase shift, evaluated to 300 as, is roughly constant for $\lambda$ between 500 and 1,000 nm. This phase shift is characteristic to the specific nanoscale dimensions of the sample, in comparison with the 500 as shift that would be expected from a macroscopic plain gold sample (Supplementary Note 6). Calculations show that the probed near-field component, normalized to the total field strength, is suppressed at the surface of the nanotaper to $\sim 0.5$ with respect to the incident laser field. The numerical simulations of the nanotaper are compared with the analytically solvable Mie calculations of an infinite cylinder with a diameter of 200 nm.

We performed trajectory calculations of the photoemitted electrons in the near fields around the nanotaper. In line with previous theoretical studies[17–20,22], the photoemission process has been assumed to be instantaneous. The retrieved streaking curves from the simulated streaking trace (Supplementary Note 6; Supplementary Fig. 6) from the tapered nanowire are shown in Fig. 3d (symbols) together with a simulated gas phase streaking curve (solid green line). The relation between the vector potential of the simulated streaking field (dashed purple line) and streaking trace (solid purple line) is shown as an inset to Fig. 3d and will be explained in the discussion section below. The streaking curve from the tapered nanowire exhibits a shift with respect to the reference streaking curve calculated using the vector potential of the incident laser (green line) of $\sim \Delta t = 300$ as with a relative amplitude of $\sim 0.4$. We also calculated the theoretical spectrally resolved response function. The spread of the theoretical values is given by the s.d. of the response averaged over different emission

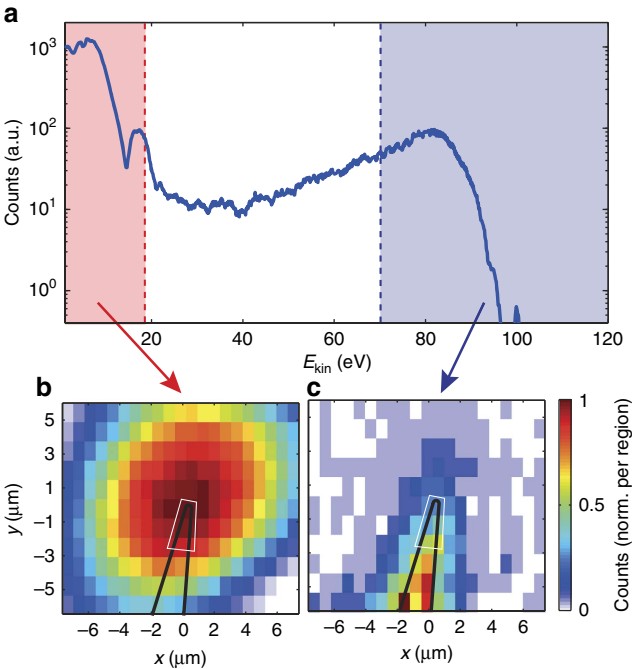

**Figure 2 | Position scan with photoelectron spectra from the Au nanotaper.** (**a**) Electron spectrum under combined illumination with XUV and NIR light, measured using a time-of-flight spectrometer. For the polycrystalline nanowire, the energy landscape under the Fermi edge results from averaging over different crystal structures and orientations, and eventually a contribution from surface contamination. (**b**) Integrated electron emission maps from the low-energy region of **a** dominated by strong-field NIR photoemission and (**c**) for the high-energy region of **a** caused by linear XUV photoemission. The laser beam propagates in the z direction. The NIR photoemission is strongly enhanced at the apex of the nanotaper, where field enhancement at the apex supports non-linear NIR photoemission processes. The XUV photoemission represents the convolution of the taper geometry with the XUV beam in the focus. The solid black line serves as a guide to the eye showing the outline of the nanotaper, while the solid white line illustrates approximately the geometry of the part of sample that is probed in the experiment. Calculations show that relative amplitude and phase of the electrical field is approximately homogenous over the probed surface region and accordingly averaging preserves results on relative phase and amplitude of the field (Supplementary Note 6; Supplementary Figs 6–8).

points, weighted by the XUV beam profile (see Supplementary Note 6; Supplementary Figs 7–9; Methods for details). The theoretical phase shift of the nanotaper streaking curve is −0.5 to −1.1 rad.

## Discussion

For our experimental parameters, the emission area on the tapered nanowire is much larger than at the taper apex, such that the streaking spectrogram is dominated by the contribution from the tapered nanowire, while streaking signals from the taper apex are not discernible. Increasing the XUV flux and improving the focusing should allow to also decipher contributions from the nanotaper apex (Supplementary Note 7; Supplementary Fig. 10).

We compared the theoretical streaking data obtained from the spatially averaged simulations to theoretical streaking curves for electrons emitted from specific positions on the tip. The curves obtained for emission at specific positions from the nanotaper (purple and light blue lines corresponding to positions 200 and 3,000 nm away from the apex of the nanotaper) are in excellent

agreement with the retrieved data from the full simulation. Between these spatial points, the theoretical 300 as shift of the nanotaper streaking curve with respect to the incident laser field is in good agreement with the experimentally measured shift of 200 ± 50 as. Comparing the experimental and theoretical spectrally resolved response functions, both the relative amplitudes and the relative phase shifts are again in good agreement. These results indicate that the experiments successfully probe the near fields around the tapered nanowire part of the sample.

To rule out other possible mechanisms for the shift between the gas phase and nanotaper streaking curves, we considered photoemission time delays and the spatial dependence of the fields around the nanotaper. While recent experimental studies compare photoemission delays of macroscopic noble metal surfaces with gaseous media for photoemission energies <30 eV (ref. 28), no other experimental studies have been reported, which compare delays between a nanostructured sample and a noble gas. In general, the photoemission process is not instantaneous, but electrons are released into the external streaking fields with some effective absolute photoemission time delay, depending also on the photoemission energy of the electrons. Theoretical studies for neon[29] suggest that the absolute photoemission delay for energies exceeding 70 eV is <10 as. In previous studies on plain macroscopic surfaces, the streaking field polarization was approximately along the surface normal[30]. This yields an additional delay, since the normal field component is screened over very short depths (in the range 0–0.3 nm for Mg (ref. 31)) at the surface, meaning that photoemitted electrons do not experience the streaking field until they reach the surface—which typically takes on the order of tens to hundreds of attoseconds[30,32–34]. In our case, however, photoelectrons from the nanotaper probe the electric field component parallel to the surface. This component is continuous across the surface, and approximately homogeneous over the electron emission depth as the skin depth (~30 nm) is significantly larger than the emission depth (~0.4 nm). Thus, delays due to the transport of the electrons to the surface are largely absent for our experiments. At the relatively high XUV energies employed in our experiments, additional effects should play a minor role (Supplementary Note 8). We therefore assume that the photoemission delay from the tapered nanowire in the given geometry is negligible compared with the measured time shift between the streaking spectrograms for the nanotaper and gas. The shift is thus attributed to the difference in the electric fields acting on the released electrons, which can be related to the collective free-electron polarization response of the gold nanowire.

Full temporal information about the nanoscale near fields is only preserved in the streaking traces if the experiment is performed in the ponderomotive streaking regime, which is characterized by a spatially homogeneous field distribution. To elucidate the streaking regime of the nanotaper near-field sampling, we calculate the adiabaticity parameter $\delta$, defined similar to strong-field photoemission experiments on localized, enhanced near fields[25] as the ratio of the time it takes an electron to leave the near field to the period of the laser pulse. Three different regimes are categorized: (i) $\delta \ll 1$, defining the 'instantaneous' regime, where the near field is probed directly, (ii) $\delta \gg 1$, which we refer to as the 'ponderomotive' regime, and the intermediate regime (iii) characterized by $\delta \sim 1$. For the parameters of the experiment, electrons take at least 10 fs to leave the near field, which is long compared with the optical period of 2.5 fs and the duration of the NIR few-cycle pulse of 5 fs. Thus, the experiment is performed in the ponderomotive streaking regime. For the given XUV photon energy, an electron propagates ~24 nm within the near fields during the NIR pulse duration of 5 fs, which is much smaller than the near-field variation length of

~200 nm from the surface as shown in Fig. 1b,c. General considerations (Supplementary Note 9; Supplementary Fig. 11) based on the adiabaticity parameter suggest that the streaking curve should only be shifted on the order of −10 as with respect to the vector potential of the near fields at the emission point. This is confirmed by simulations, revealing a minor contribution of −20 as (Fig. 3d), which is small compared with the measured shift (Supplementary Fig. 12). The measured streaking curve thus provides direct access to the temporal evolution of the vector potential and consequently also the electric near fields around the nanotaper through equation 1.

In conclusion, we have successfully implemented ANNS on Au nanotapers. By theoretically analysing the photoelectron trajectories and the spatial homogeneity of the near fields, we were able to directly access the local near fields on nanometre spatial and attosecond temporal scales. Our measurements show that the near fields are shifted by (200 ± 50) as with respect to the incident laser field, in good agreement with the theoretical shift of 300 as. Our approach paves the way towards studying more complex

structures, including the characterization of ultrafast nanocircuits. Understanding the local near fields, for example, in simple optical-field-driven switches, will allow constructing more complex, coupled nanostructures that may lead to the initial building blocks for petahertz electronics. Spatial resolution can be achieved by scanning the XUV beam (with a smaller focus size) over the surface, or by using a photoemission electron microscope.

## Methods

**Experimental scheme for ANNS.** Few-cycle NIR laser pulses (4.5 fs centred at 720 nm) are produced using a Ti:Sa laser system (Femtopower Compact Pro, Femtolasers) together with a hollow core fibre for spectral broadening and subsequent recompression by chirped mirrors. The vacuum set-up used in our experiments is described in more detail in Supplementary Note 1 and Supplementary Figs 1 and 2. In brief, focusing the few-cycle pulses into Ne gas generates high harmonic radiation. The high harmonics are inherently synchronized in time to the NIR laser pulses that generated them. The high harmonics and NIR are spatially and spectrally separated into two annular parts using a 0.15-µm-thick Zr foil mounted on a pellicle, fabricated from nitrocellulose. The co-propagating laser pulses are focused (focal length f = 12.5 cm) onto a nanoscopic sample (wet-etched Au taper, 100-nm apex radius and opening angle of 10°) using an annular double mirror, the inner part of which is mounted on a piezo delay stage, introducing a variable attosecond delay between the XUV and NIR pulses. This mirror also spectrally selects a 7 eV bandwidth of high harmonics centred at 95 eV to produce an isolated 220 as pulse. We define the zero time delay between the pulses to be the time, at which the maxima of the intensity envelopes coincide. The streaking spectrograms in Fig. 3 were recorded by using a TOF spectrometer to measure the photoelectron spectrum, as a function of the time delay between the two pulses.

Three-dimensional motorized stages facilitate positioning the taper with nanometre precision. In addition to the nanotaper target, the stages allow to position a gas nozzle in the laser focus offering the possibility to characterize the NIR-field and the XUV-pulse parameters by recording a reference streaking trace in neon. Photo-excited electrons are propagating through the near field of the nanotaper and the NIR-laser field, the latter polarized parallel to the taper axis, accelerating or decelerating the electrons on their way to the TOF. The TOF of the electrons is recorded and afterwards converted to kinetic energy. Due to the vanishingly small count rates at high energies of <0.1 counts per shot, a lens voltage of 500 V was applied to the TOF to enhance the high-energy counts.

Compared with the gas streaking, electron count rates are more than one order of magnitude lower from the nanotaper because of the nanoscale size of the sample. This leads to the overall acquisition times of up to a few hours to obtain reasonable statistics of the streaking trace. The acquisition time is mainly limited by the time the carrier-envelope phase (CEP) can be locked. To exclude the phase drifts and instabilities during measurements, several nanotaper and gas streaking spectrograms are recorded, compared and, for stable conditions, superimposed.

In contrast to gas phase streaking, the expanded density of states of the Au nanotaper target leads to severe broadening of the energy spectrum, resulting in streaking curves with a width of >20 eV. Inelastic scattering of photoelectrons on

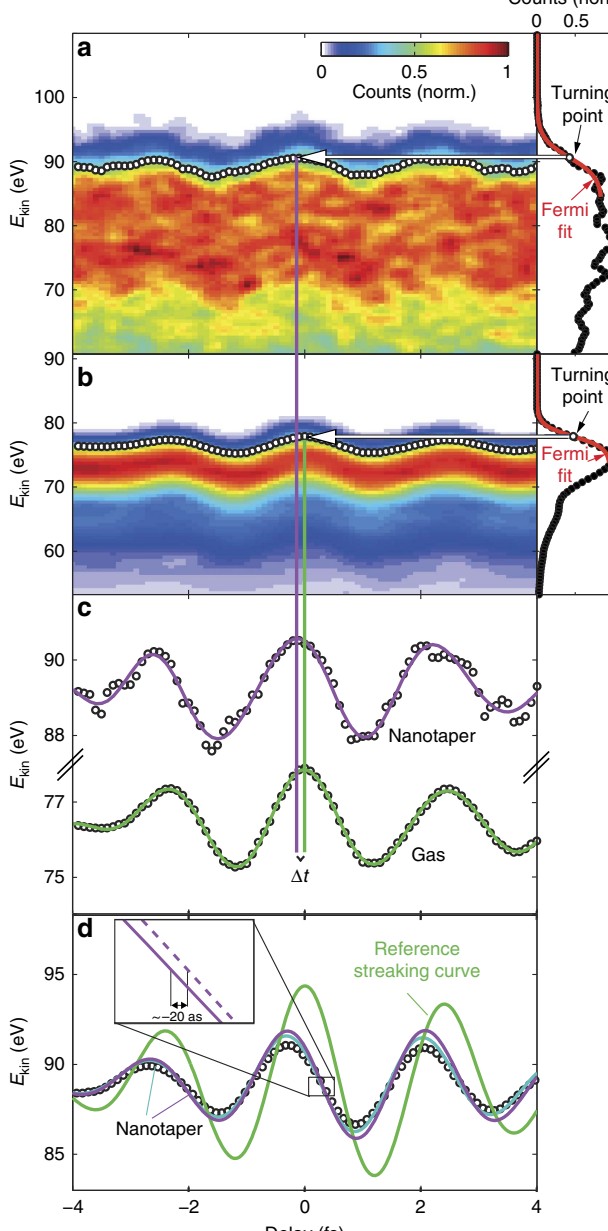

**Figure 3 | Analysis of delay shifts between streaking of a tapered nanowire and gas.** Measured data for (**a**) the Au nanotaper and (**b**) Ne. The right panels of the spectrograms show electron spectra for a fixed delay of −0.2 fs (nanotaper) and 0 fs (gas) illustrating extraction of the streaking curves. A Fermi function (red) is fitted to the cutoff edge of the spectrum, since the high-energy part of the spectrum is exclusively determined by gold. The turning points of the Fermi functions for different delay times provide the curves depicted by symbols in **a** and **b** (Supplementary Note 3; Supplementary Figs 4 and 5). The fine structure in the nanotaper streaking spectrograph in **a** results from experimental noise, which is predominantly from counting statistics. Typically gas streaking spectra were recorded with count rates of ~2 counts per laser shot, while tip streaking spectra were recorded with count rates of <0.1 counts per laser shot. (**c**) The retrieved curves are smoothed by Fourier filtering (solid lines) allowing to determine the shift $\Delta t$ between them for every delay. (**d**) The streaking curve retrieved from a Monte Carlo simulation (symbols; Methods; Supplementary Note 4). The purple and light blue lines illustrate streaking curves for electrons emitted from the front of the nanowire at y = −200 nm and y = −3,000 nm, respectively. The solid green line shows the streaking curves from the reference in neon gas. To aid comparison of the streaking curves, the reference gas streaking trace was upshifted in energy to the streaking traces from the gold tip. The inset shows the relation between the simulated streaking curve (solid line) and the local vector potential of the near field (dashed line) at the emission point.

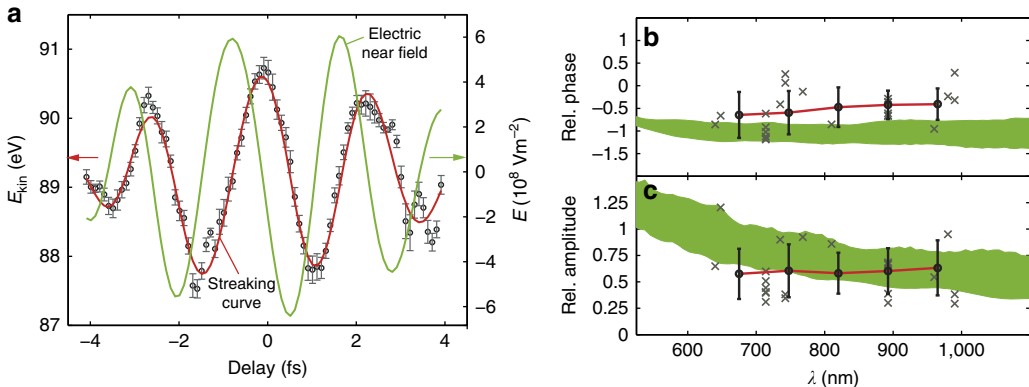

**Figure 4 | Extraction of the electric near field and response function.** (**a**) Reconstruction of the local electric near field (green line) and vector potential (red line) at the nanotaper surface from the measured streaking curve (symbols). The error bars indicate 95% confidence intervals of the Fermi fit. Retrieved wavelength-resolved (**b**) relative phase shift and (**c**) amplitude obtained from different measurements. Data points are shown as crosses, whose positions are given by the spectral sampling of the different measurements. Different measurement sets do not necessarily cover the same number and size of delay scans in the streaking spectrographs, which translates into a different spacing and number of data points after Fourier transformation. Only data points in the wavelength range covered by the input NIR pulses, that is, from 500 to 1,100 nm, are displayed. Data points outside this range are a consequence of over-sampling in the time domain, and have a vanishingly low spectral amplitude. Mean values (symbols) have been obtained by linearly interpolating the retrieved linear response from the individual measurements. Error bars represent the s.d. The green shaded areas show the range of phase shifts and relative amplitudes expected from the electric field calculations, considering emission from different points on the surface from the tapered nanowire.

their way to the surface additionally broadens the spectra. Irrespective of these effects, analysis of electrons with kinetic energies close to the high-energy cutoff of the XUV photoemission provides access to the near-field dynamics at the surface of the Au nanotaper.

**Nanotaper preparation.** Gold nanotapers are produced from 0.1-mm-thick poly-crystalline Au wires in a lamella drop-off technique by wet electrochemical etching using 90% saturated KCl as etchant. Using this method, apex radii between 20 and 100 nm can be obtained with an opening angle of typically 10°. Excellent surface quality with roughness <0.8 nm can extend up to 300 μm from the apex downwards[35].

**Simulation details.** The optical near fields are calculated using the FDTD approach for a nanocylinder with a radius of 100 nm and half sphere at the apex. NIR fields are simulated by a Gaussian pulse centred at 720 nm and a duration of 4.5 fs (full-width at half-maximum, FWHM). The attosecond near-field streaking process was modelled similar to ref. 19, assuming an XUV spot size of 5 μm (FWHM) and an IR laser intensity of $10^{12}$ Wcm$^{-2}$. For the trajectory simulations using a Monte Carlo approach for retrieval of the streaking spectrogram, electrons are initialized at random positions on the nanotaper surface, by projecting the spatial profile of the XUV beam on the nanotaper. Random initial energy is given by the experimental XUV photoelectron spectrum. The electron emission time distribution is given by the XUV pulse duration (220 as FWHM), while the emission angle distribution is assumed to be isotropic. Subsequently to their photoemission, electrons are propagated classically in the electric near fields around the nanotaper. Only electrons having a final propagation direction within the detection angle of the TOF (22.5° with respect to the taper axis) are recorded. The streaking curves from selected emission points were calculated assuming a fixed initial energy and emission angle, and neglecting the finite duration of the XUV pulse.

The expected response function in Fig. 4b,c is obtained by calculating the electric fields with the same approach as above using a tapered nanowire with an opening angle of 10° and 50 nm radius of the hemisphere. The response is averaged over $10^4$ points on the surface, weighted by the XUV beam profile. The width corresponds to the s.d.

**Electric field and response function reconstruction.** In the ponderomotive streaking regime, the electric fields can be approximated as homogeneous and the final change of momentum $\Delta \mathbf{p}$ of the electrons emitted at time $t_0$ can directly be related to the component of the vector potential **A** parallel to the emission direction: $\Delta \mathbf{p} = -e\,\mathbf{A}(t_0)$ (in the Coulomb gauge). This allows a direct reconstruction of the local electric field $\mathbf{E}(t)$ from the measured shift $\Delta E_{kin}$ in the kinetic energy of the electrons recorded within the streaking spectrogram:

$$\mathbf{E}(t_0) = \frac{1}{2e} \frac{\sqrt{2m_e}}{\sqrt{(E_0 + \Delta E_{kin})}} \frac{\partial \Delta E_{kin}(t_0)}{\partial t_0} \quad (2)$$

where $e$ is the elementary charge, $m_e$ the mass of the electron and $E_0$ is the initial kinetic energy. The values $\Delta E_{kin}$ and $E_0$ are obtained from the extracted streaking curve. The Fourier-filtered curves allow a direct calculation of the derivative and reconstruction of the amplitude and phase of the electric field in a delay-dependent

manner. The amplitude of the incident laser electric field obtained from the gas streaking measurements is typically underestimated from gas measurements. For correction factors, the field amplitudes from gas measurements are compared with the expected field amplitudes from the cutoff energies of direct strong-field electron emission (Supplementary Note 4). The response function is retrieved by discrete Fourier transforming the complex electric fields reconstructed from the gas and nanowire measurements, and comparing amplitude with phase in the spectral domain.

**Data availability.** The data that support the findings of this study are available from the corresponding authors upon request.

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

## Acknowledgements

We acknowledge the co-workers that have helped to build the attosecond infrastructure, in particular Thorsten Uphues and Adrian Wirth. We are grateful to Ulf Kleineberg for supporting the fabrication of multilayer extreme ultraviolet mirrors and to Seungchul Kim for fruitful discussions. We are grateful for support by the Max Planck Society and the German Research Foundation (DFG) through SPP1391, and the Cluster of Excellence: Munich Centre for Advanced Photonics (MAP). B.F. acknowledges support from Marco Allione and Enzo Di Fabrizio via the King Abdullah University of Science and Technology (KAUST). F.S., S.Z. and M.F.K. acknowledge support from the European Union (EU) via the European Research Council (ERC) grant ATTOCO, M.K., M.F. and P.H. via the ERC grant NearFieldAtto. This research has also been supported in part by the Global Research Laboratory program (grant no. 2009-00439), by the Leading Foreign Research Institute Recruitment program (grant no. 2010-00471) and by the Max Planck POSTECH/KOREA Research Initiative (grant no. 2011-0031558) through the National Research Foundation of Korea (NRF). For M.I.S. research, the main support came from grant no. DE-FG02-11ER46789 from the Materials Sciences and Engineering Division of the Office of the Basic Energy Sciences, Office of Science, US Department of Energy, and an additional support was provided by grant no. DE-FG02-01ER15213 from the Chemical Sciences, Biosciences and Geosciences Division, of the Office of the Basic Energy Sciences, Office of Science, US Department of Energy.

## Author contributions

B.F., J.S. and F.S. contributed equally to this work. M.F.K. and P.H. conceived the experiment. B.F., J.S., F.S., K.W. and B.A. performed the measurements. A.K., S.Z. and S.A.T. helped with the laser operation and optimization. M.F. and M.K. prepared and characterized the tapered Au nanowires. A.G. and V.P. prepared the specialized optical components. J.S. and F.S. developed the Monte Carlo simulation model and performed the simulations. B.F., J.S., F.S., M.F., W.A.O., A.M.A., D.K., P.H., F.K. and M.F.K. evaluated, analysed and interpreted the results. All authors discussed the results and contributed to the final manuscript.
