## [Peer Review File · Nature Communications]

Reviewers' Comments:

Reviewer #3 (Remarks to the Author)

The manuscript describes attosecond streaking experiments performed on a solid-state nanoscale system--a gold nano-tip. The experiments constitute the first extension of attosecond streaking spectroscopy to such a system and are therefore both novel and interesting. Due to the relatively large size of the focused XUV beam and the linear nature of the XUV absorption process, the authors are unable to probe the localized near-field at the apex of the gold tip. However, through careful analysis, the authors are able to estimate the temporal profile of the near-field surrounding the tip's shank and extract estimates of the spectral response of this near-field to NIR laser excitation. Although similar probing of the localized and enhanced field near the tip's apex would be of significantly greater interest, these results do constitute the first demonstration of attosecond streaking at such a nanostructure and highlight challenges associated with this technique. Altogether, I find the manuscript acceptable for publication in Nature Communications after the following concerns are addressed (organized by section of the paper).

1. Abstract / Introduction

a) The idea of 'attosecond nanoscale near-field sampling' suggests sampling in both time (attosecond) and space (nanoscale). In these experiments, there is temporal locality, i.e. sampling in the time-domain, but there is not spatial locality, i.e. the results are ultimately averaged over the larger-than-nanoscale area of the XUV beam and nano-tip's overlap region. Given this fact, I suggest changing 'attosecond nanoscale near-field sampling' in the abstract back to 'attosecond near-field retrieval' (from an earlier version of the manuscript).

b) The first paragraph of the introduction requires changes. Firstly, Ref. 11 is misplaced. This reference does not relate to femtosecond metrology of nanolocalized fields (it was erroneously suggested in an earlier round of review). Secondly, in the last sentence of the first paragraph of the introduction, the concept of 'charge interactions' is vague (all other elements in the list are forms of charge interactions). This phrase should be clarified or removed.

c) The paragraph describing ANNS (second paragraph of the introduction) requires editing. Firstly, the phrase "...that is subsequently accelerated in the near-fields" is a misplaced modifier; this sentence must be rearranged. Secondly, I find the following sentences misleading: "...XUV photoemission results in high-energy electrons being emitted from the entire illuminated area that is typically not confined to a nanoscale but rather a micron scale. Consequently, it is challenging to characterize spatially varying near-fields". If the XUV illuminates an isolated nano-object, then this is not a concern. In fact, most of the referenced work on theoretical investigations of attosecond streaking spectroscopy near solid-state nanostructures (Refs. 18-23) explore such a scenario. It should be made clear that this limitation applies to the particular nano-tip geometry used here. Lastly, I do not understand the comment regarding non-linear effects. From context, this seems to refer to non-linear effects associated with the XUV illumination, but under what exotic circumstances would non-linear effects occur with the XUV pulse? Does this concern really need to be addressed?

2. Results

a) In the discussion of Figs. 1 and 2, several adjustments should be made. First, the size of the cylinder used in the analytical Mie calculations should be noted. Additionally, the paper should direct the reader to the Supplementary Information where it is relevant. For example, the relevant sections of the Supplementary Information should be pointed to following the comments: "...the end of the hemisphere... does not yield any detectable signal above the noise level..."; in the caption for Fig. 1, "Slight position dependent oscillations occur due to a plasmon launched at the tip apex"; and lastly, in the caption for Fig. 2, "Calculations show that the relative amplitude and phase of the electrical field is approximately homogeneous...".

b) The fine-structure under the streaking curve presented in Fig. 3a is attributed to counting statistics (in the caption for Fig. 3). Please provide a comparison in the electron count rate per laser shot for the nano-tip versus the gas experiments. Additionally, the reference streaking curve provided in Fig. 3d comes from the neon experiments. Please explicitly state that in Fig. 3d this curve has been shifted up in energy to overlap with that from the nano-tip.

c) The presentation of Fig. 3d is confusing. Fig. 3 is first discussed in the 'Results' section, but Fig. 3d is not discussed. Components of Fig. 3d are later explained in the second paragraph of the 'Theoretical Results' section, but the dashed purple curve is still not explained. Finally, the dashed purple curve is explained in the second to last paragraph of the 'Discussion' section. Overall, the readability of the paper would be greatly improved by combining the 'Results' and 'Theoretical Results' section, and by indicating that a component of a figure that is not discussed will be discussed later. As it stands now, each figure is discussed in the 'Results' section. Then discussed again in the 'Theoretical Results' section and then once more in the 'Discussion' section.

d) The data presented in Fig. 4b needs clarification. The authors state that this data follows from a discrete Fourier transform of the reconstructed electric field at the nano-tip. However, the reconstructed electric field in Fig. 4a consists of what appears to be on the order of a hundred or so points while Fig. 4b shows on the order of ten different wavelengths. Does this spectral response data derive from different streaking data that involves electric field reconstructions with only around ten points? If so, what are the limitations associated with having so few samples in these streaking experiments? How does the sampling rate used to acquire only ten or so points compare to the expected required Nyquist frequency for reconstructing the nano-tip field? It seems it should be sufficient due to the approximately two-cycle nature of the pulse, but a comment is merited. Further details and explanation regarding exactly where this data comes from are required in the main text or the Supplementary Information.

Response to the Referee:

1. Abstract / Introduction

a) *The idea of 'attosecond nanoscale near-field sampling' suggests sampling in both time (attosecond) and space (nanoscale). In these experiments, there is temporal locality, i.e. sampling in the time-domain, but there is not spatial locality, i.e. the results are ultimately averaged over the larger-than-nanoscale area of the XUV beam and nano-tip's overlap region. Given this fact, I suggest changing 'attosecond nanoscale near-field sampling' in the abstract back to 'attosecond near-field retrieval' (from an earlier version of the manuscript).*

As recommended by the referee we changed the wording from 'attosecond nanoscale near-field sampling' to 'attosecond near-field retrieval'. The changes to the manuscript are shown in red.

b) The first paragraph of the introduction requires changes. Firstly, Ref. 11 is misplaced. This reference does not relate to femtosecond metrology of nanolocalized fields (it was erroneously suggested in an earlier round of review). Secondly, in the last sentence of the first paragraph of the introduction, the concept of 'charge interactions' is vague (all other elements in the list are forms of charge interactions). This phrase should be clarified or removed.

As pointed out by the referee, Ref. 11 is misplaced since it refers to nonlinear optical spectroscopy, rather than femtosecond metrology, of nanolocalized fields. We therefore deleted the reference, as recommended by the referee.

We agree that the term 'charge interactions' is vague, and we have removed the phrase from the manuscript.

c) The paragraph describing ANNS (second paragraph of the introduction) requires editing. Firstly, the phrase "...that is subsequently accelerated in the near-fields" is a misplaced modifier; this sentence must be rearranged.

We thank the referee for pointing out that the modifying phrase "...that is subsequently accelerated in the near fields" can be misinterpreted as referring to the XUV optical pulse. To make it unambiguous that it is the photoelectrons that are accelerated, we changed the sentence as follows:

"... ANNS relies on the emission of photoelectrons with high initial momentum by an attosecond extreme ultraviolet (XUV) pulse, and subsequent acceleration of the photoelectrons in the near-fields."

Secondly, I find the following sentences misleading: "...XUV photoemission results in high-energy electrons being emitted from the entire illuminated area that is typically not confined to a nanoscale but rather a micron scale. Consequently, it is challenging to characterize spatially varying near-fields". If the XUV illuminates an isolated nano-object, then this is not a concern. In fact, most of the referenced work on theoretical investigations of attosecond streaking spectroscopy near solid-state nanostructures (Refs. 18-23) explore such a scenario. It should be made clear that this limitation applies to the particular nano-tip geometry used here.

We improved the clarity of this part of the text by changing the wording (indicated in the main text by red color) as follows:

“... As a linear process, XUV photoemission results in high-energy electrons being emitted from the entire illuminated area that is typically not confined to a nanoscale but rather micron scale. Consequently, the detection scheme averages on the micron scale making it challenging to characterize near-fields around nanostructured sample geometries typically varying substantially in phase and amplitude on a nanometer scale.¹⁷⁻²⁰ Here, we perform ANNS measurements on a nanotaper at near-infrared intensities well below the onset of non-linear effects. Using the gold nanotaper sample geometry, we show that through careful analysis of field homogeneity and streaking electron trajectories a meaningful attosecond characterization of near-fields can be performed in spite of the inherent challenges associated with the large emission area. ...”

However we want to stress that the characterization of nanoscale electrical near-fields in nanoscale sample geometries is not as straightforward as in gas phase streaking. Electric near-fields were found to vary substantially in amplitude and phase on scales smaller than the nanosample. The XUV focus size leads to photoemission over the whole nanosample geometry, thereby averaging over different near-field phases and amplitudes and producing a more complicated streaking trace than for gas phase measurements. As an example, in reference 18, numerical calculations for a spherical geometry were performed showing the smearing out of streaking traces. Subsequently near-field properties such as phase and amplitude are indiscernible. Similar theoretical calculations were performed for different nanosample geometries with similar results (see. Ref 17, 19-20). The main challenge is extracting meaningful near-field characterization parameters at a particular spatial region of the sample, from a measurement encompassing a larger spatial scale over which these parameters vary.

Lastly, I do not understand the comment regarding non-linear effects. From context, this seems to refer to non-linear effects associated with the XUV illumination, but under what exotic circumstances would non-linear effects occur with the XUV pulse? Does this concern really need to be addressed?

We were referring to nonlinear effects (such as metallization) from the near-infrared pulse, not from the XUV pulse. To clarify this, we have amended the first sentence in the last paragraph of the introduction to:

“Here, we perform ANNS measurements on a nanotaper at near-infrared intensities well below the onset of non-linear effects ...”

2. Results

a) In the discussion of Figs. 1 and 2, several adjustments should be made. First, the size of the cylinder used in the analytical Mie calculations should be noted. Additionally, the paper should direct the reader to the Supplementary Information where it is relevant. For example, the relevant sections of the Supplementary Information should be pointed to following the comments: "...the end of the hemisphere... does not yield any detectable signal above the noise level..."; in the caption for Fig. 1, "Slight position dependent oscillations occur due to a plasmon launched at the tip apex"; and lastly, in the caption for Fig. 2, "Calculations show that the relative amplitude and phase of the electrical field is approximately homogeneous...".

We followed the referee's recommendation and added the size of the infinite cylinder that was used in the Mie calculation (see. Fig 1 d) and e)). The first paragraph of the 'Theoretical Results' section now reads:

“...The numerical simulations of the nanotaper are compared to the analytically solvable Mie calculations of an infinite cylinder with a diameter of 200 nm. ...”

In addition the Figure caption to Fig. 1 now reads:

“...at the nanotaper showing the absolute value and phase dependence on the wavelength. The response of the shank is close to an infinite cylinder with a diameter of 200 nm calculated using Mie theory (black dashed line). ...”

We added a direct reference to the important Supplementary section in the main text, and Figure caption 1 and Figure caption 2, as suggested by the referee (changes are indicated in the manuscript).

b) The fine-structure under the streaking curve presented in Fig. 3a is attributed to counting statistics (in the caption for Fig. 3). Please provide a comparison in the electron count rate per laser shot for the nano-tip versus the gas experiments. Additionally, the reference streaking curve provided in Fig. 3d comes from the neon experiments. Please explicitly state that in Fig. 3d this curve has been shifted up in energy to overlap with that from the nano-tip.

We added the count rates for streaking in gas, and on tips. The caption to Figure 3 now reads:

“The fine structure in the nanotaper streaking spectrograph in a) results from experimental noise, which is predominantly from counting statistics. Typically gas streaking spectra were recorded with count rates of approximately 2 counts per laser shot, while tip streaking spectra were recorded with count rates of less than 0.1 counts per laser shot.”

The reference streaking trace measured in Neon gas and shown in Fig. 3d) was upshifted in energy to aid comparison with the streaking trace from the tapered nanowire. To make this clear, we amended the caption to Figure 3 as follows:

“...The solid green line shows the streaking curves from the reference in neon gas. To aid comparison of the streaking curves, the reference gas streaking trace was upshifted in energy to the streaking traces from the gold tip. ...”

c) The presentation of Fig. 3d is confusing. Fig. 3 is first discussed in the 'Results' section, but Fig. 3d is not discussed. Components of Fig. 3d are later explained in the second paragraph of the 'Theoretical Results' section, but the dashed purple curve is still not explained. Finally, the dashed purple curve is explained in the second to last paragraph of the 'Discussion' section. Overall, the readability of the paper would be greatly improved by combining the 'Results' and 'Theoretical Results' section, and by indicating that a component of a figure that is not discussed will be discussed later. As it stands now, each figure is discussed in the 'Results' section. Then discussed again in the 'Theoretical Results' section and then once more in the 'Discussion' section.

We wish to show Figure 3(d) as part of Figure 3 to aid comparison between experimental and theoretical results. We also want to make clear the important distinction between the experimental results and theoretical results by subdivision of the “Results” section, and therefore would prefer not to combine these into a single section. In our opinion, the current layout is the clearest way to present the work. To improve the clarity of the text referring to Figure 3 (d) slightly, we made the following change to the text:

“...curves from the simulated streaking trace (see Supplementary Note 5 and Supplementary Fig. 6) from the tapered nanowire are shown in Fig. 3(d) (symbols) together with a simulated gas phase streaking curve (solid green line). The relation between vector potential of the simulated streaking field (dashed purple line) and streaking trace (solid purple line) is shown as inset to Fig 3(d) and will be explained in the discussion section below. The streaking curve from the tapered nanowire exhibits a shift... “

d) The data presented in Fig. 4b needs clarification. The authors state that this data follows from a discrete Fourier transform of the reconstructed electric field at the nano-tip. However, the reconstructed electric field in Fig. 4a consists of what appears to be on the order of a hundred or so points while Fig. 4b shows on the order of ten different wavelengths. Does this spectral response data derive from different streaking data that involves electric field reconstructions with only around ten points? If so, what are the limitations associated with having so few samples in these streaking experiments? How does the sampling rate used to acquire only ten or so points compare to the expected required Nyquist frequency for reconstructing the nano-tip field? It seems it should be sufficient due to the approximately two-cycle nature of the pulse, but a comment is merited. Further details and explanation regarding exactly where this data comes from are required in the main text or the Supplementary Information.

As stated by the referee, in a discrete Fourier transformation of a discrete time signal, the number of data points in frequency space and time space is equal. Therefore, for the experimental data presented in the manuscript, the number of data points in the spectral domain, shown in Fig 4b), must be equal to the number of data points shown in the temporal domain, shown in Fig 4a). In Fig 4b) the spectral domain is only shown from 500 nm to 1100 nm, which is approximately the wavelength range covered by the NIR pulses used in the experiments. Not all data points lie within this wavelength range because the electric field in the time domain is sampled at a much higher rate than the Nyquist frequency. However, the spectral amplitude of data points outside the range 500 nm to 1000 nm is vanishingly low.

The spectral response is not derived for different streaking traces involving only about 10 data points as stated by the referee. In total 10 streaking measurement sets are included in the evaluation of Figs. 4b) and c). The shortest data set covers a delay range of 6.5 fs with a step size of 150 as and the longest data set covers a delay range of 9 fs with a step size of 100 as. All other measurement sets use intermediate delay ranges and step sizes between these values.

To clarify this in the manuscript, we added the following sentence in Figure caption 4:

“...into a different spacing and number of data points after Fourier transformation. Only data points in the wavelength range covered by the input NIR pulses, i.e. from 500 nm to 1100 nm, are displayed. Data points outside this range are a consequence of over-sampling in the time domain, and have a vanishingly low spectral amplitude. Mean values ...”.